# Product-Service System Business Modelling Methodology Using Morphological Analysis

**Minkyu Kwon [1], Jihwan Lee [2],\* and Yoo S. Hong [1]**

[1] Department of Industrial Engineering and Institute for Industrial Systems Innovation, Seoul National University, Seoul 08826, Korea; mkkwon@snu.ac.kr (M.K.); yhong@snu.ac.kr (Y.S.H.)

[2] Division of Systems Management and Engineering, Pukyong National University, Busan 48513, Korea

\* Correspondence: jihwan@pknu.ac.kr; Tel.: +82-51-629-6492

**Abstract:** This study aims to establish a structured business modelling methodology for successful implementation of Product-service system (PSS). The morphological analysis is applied in investigating the possible patterns of the PSS business model. To systematically collect the business model patterns, we have collected a set of predefined building blocks which can be used in business modelling. These building blocks are collected through investigation of actual PSS business model cases. By mixing and matching various building blocks, various innovative business model alternatives can be designed. To demonstrate our morphological chart, real case example of hair dryer company is illustrated. Moreover, we introduce a web-based system, which supports our business model idea generation procedure using morphological chart.

**Keywords:** business model; business modelling; morphological analysis; morphological chart; Product-Service System (PSS); case-based system

---

## 1. Introduction

Reduced profit rate due to fierce competition and rapid changes in customer expectations have forced manufacturing companies to shift from the sales of physical products to the sales of functionality delivered by the integration of the product and related services [1,2]. This novel concept is referred to as Product-service System (PSS). By the provision of integrated service along with the product life-cycle, PSS could enhance the Long-term profit by increasing the interaction between customer and companies [3]. PSS also could achieve an improved utilization and reduced environmental impact by accessing customer usage data and providing customized service for each customer [4].

Despite of its economic and environmental potential, shifting towards PSS is challenging [5] because it involves radical transformation of value chain and organization [6]. Kuo et al. [7] and Gebauer et al. [8] confirms that many of manufacturer do not deliver PSS effectively, due to poorly defined customer requirement and service portfolios. In this regards, literature emphasized that planning of sound business model might be a critical factor for successful PSS employment [9–12]. Business model is defined as a logical description of how a company will generate revenue and make a profit with its product and service [13]. Even with the same product, a company could differentiate their business model with different configuration of its revenue model, distribution channel or customer segmentation and so forth. Literature shows that in some cases a company's choice on the business model may be more important than the product design [14] and business model innovation plays a critical role in overcoming firm's facing challenges and in preparing the industry turbulence of the future [15]. Due to its importance, literature attempted to clarify the concept of the business model [16–19]. However, there is lack of consensus on the general characterization or classification scheme about the business model [20–22].

Business model also has not been discussed extensively in PSS literature [23,24]. As Tukker and Tischner [10] noted, the current PSS researches focus too much on individual case studies and relatively little attention has been made to support business model planning. Although there exist few BM frameworks that highlights viewpoints to understand the PSS business model (e.g., Barquet et al. [12]; Beuren et al. [25]), they are still lacking more fine-grained identification of patterns or components which can facilitate ideation of PSS business models [26]

To bridge the research gap, this study proposes a procedure for PSS business model planning using morphological analysis. Morphological analysis, which was originally proposed by Zwicky [27], is a problem solving approach that hat splits a product/solution into smaller dimensions that can then be analysed and ideated for independently. Afterwards those ideas can be mixed and matched to develop different solutions. Morphological analysis has been widely used in engineering design domains [28]. Especially, in conceptual design phase, morphological analysis is used in concept generation phase, because it allows systematic exploration of possible concept alternatives [29]. Following the same procedure, we have also decomposed the PSS business model into several perspectives and identifies possible patterns of each perspective. Each BM pattern was obtained and verified by the investigation of actual PSS business model cases. As a result, a morphological chart of PSS business model that can be used to generating various business model ideas was obtained. In order to demonstrate our procedure, a case example of hair dryer company in South Korea is introduced. The case study shows that our procedure can be used to generate variety of interesting business model alternatives. Also, we developed a web-based system, BizChef, that enables a user to navigate various business model cases and to develop a new business model easily.

The remainder of this paper is organized as follows. In Section 2, relevant literature about business model and contribution of our study is discussed. In Section 3, a detailed procedure for developing our business model morphological chart is depicted. In Section 4, a case example of business modelling using our morphological chart is illustrated. In Section 5, the prototype system, BizChef, is introduced. Finally, chapter 6, future works and conclusion of our work is discussed.

## 2. Literature Review

The interest in business models has seen continuous growth with the advent of and the explosion in the number of online companies [16]. In the new environment where the firm should deliver new information services that users often expect to receive without charge [20], firms achieve great success or suffer from failure depending on the changes in their ways of doing business. Observing these changes, the notion of the business model has received lots of attention as a tool for explaining the way of how businesses deliver value to the customer and how they can capture value from providing products and services. To uncover its concept and nature, the literature has proceeded from two approached: (1) taxonomy-based approach; and (2) model-based approach.

The objective of the taxonomy-based approaches is to develop a classification scheme of BM by investigating existing business model cases. One of the first attempts was that of Timmers [16]'s who developed the taxonomy of the internet business model. He identified the eleven types of business model on the web, such as e-shop, e-auction and third-party. Focusing on the types of revenue model, Afuah and Tucci [30] proposed the different classification scheme for internet business. Their taxonomy includes seven types of revenue model: commission, advertising, mark-up, production, referral, subscription and fee-for-service. Rappa [17] refined the categories considering the nature of value proposition as well as the revenue model. Linder and Cantrell [31] developed a more generalized taxonomy which includes other types of business besides the Internet domain. They classified a certain business model according to two criteria; the firm's core activity including selling, doing channel role or management service and relative position on the continuum of price and value from premium price and high value offerings to low price and standardized value ones. Chen et al. [32] divided concepts of equipment maintenance business model innovation into two types based on 5 classification criteria respectively and described the connotation of them by combining functions of the equipment

maintenance services. Although the taxonomy-based approach provides a good knowledge about emerging business patterns, it is limited in its applicability because it remains at the scope of specific industries and is based only on ex-post cases hence provides no guide in ideation.

The objective of model-based approaches is to propose a logical model of a BM by identifying its architecture or components. Petrovic et al. [18] defined a business model as a system which consists of resources and activities interacting with outbound environment such as customers, competitors and technologies. Morris et al. [19] defined a business model as a set of decision variables of three different layers; generic strategic decisions, tactical decisions and actual quantifiable variables. In addition to this, they also provided predefined set of such decisions which could be used in representing a business model. Zott and Amit [33] proposed the activity based model, which insisted that configurations of activities determined a business model. Alternatively, Mahadevan [34] highlighted the relationship between stakeholders as a major component of a business model. According to him, three different kinds of stream—value, revenue and logistic—between stakeholders determined a firm's business model. Alt and Zimmermann [35] proposed the six generic elements which were commonly mentioned in other literature; mission, governance structure, process, revenue sources, technology and legal issue. Osterwalder [36] developed comprehensive ontologies of the business model. His ontology consisted of basic components along with the identification of all the possible configurations and relationship between components. Although his ontology enables a structured analysis on a business model in a comprehensive manner, the complexity of entire ontology makes it hard for planners to understand the whole picture of a business model.

Business modelling issue has been addressed in some PSS literature. Lay et al. [37] and Gaiardelli et al. [38] proposed a typology of service elements that can be offered by the PSS. However, these approaches only focus on business-to-business context and cannot be applied to business-to-customer context. Kim et al. [39] developed a case-based PSS idea recommendation system which could be applied to more generic business context. Bocken et al. [40] investigated numerous business models related to environmental sustainability. Reim et al. [9] systematically classified PSS literature and investigated business model-related topics and possible tactics which could be considered in implementation of PSS business model. Mattos and Albuquerque [41] empirically deduced 6 business models in circular context by identifying success factors and strategies and linking them from case studies. Although these literature shows various approaches for defining or representing the business model by establishing its taxonomy, constituent architecture or representation model, none of them mainly deals with the structured procedure for business modelling.

Some researchers have applied morphological analysis to derive business model ideas in a systematic manner. For example, Seidenstricker et al. [42]. proposed a simple business model morphological matrix which consists of six generic fields and its belonging design options. Im and Cho [43] proposed case based reasoning methodology utilizing business model morphological chart. In terms of PSS design, Haber et al. [44] and Harber and Fargonoli [45] proposed design procedures that utilized morphological analysis to generate PSSs at a conceptual level. Utilizing morphological analysis, our work is also aligned with above studies. However, our morphological analysis focuses on generating ideas for planning PSS business model. Also, our PSS business model morphological chart was obtained throughout the comprehensive investigation of actual business cases, therefore, it is generic and balanced enough to generate various business model ideas. In the next chapters, our procedure is introduced in more detailed manner.

## 3. Business Model Strategy Identification Using Morphological Analysis

The objective of this research is to collect reusable business model patterns in order for them to be recombined to create a new business model. Investigating many business model cases, we have found that a business model pattern that is common in one industry could be an innovative solution to the other industry. For example, advertisement fee is a common revenue model of internet industry. They provide free service or contents to the users instead charges advertisers who would be attracted

by a larger base of free users [46]. Nowadays, advertisement revenue model is also found in offline business. 'Tada copy,' which is a paper copy shop in Japan, designed its revenue structure that puts advertisements on the other side of the copy papers instead of providing the copy service to students for free [47]. As shown with this example, numerous interesting business models have emerged by combining patterns of other business models [31].

In order to systematically collect business model patterns, the morphological analysis is applied. Morphological analysis is a structured problem solving methodology, which decomposes the problem or subject into a number of fundamental dimensions that can describes it as complete as possible and various features it can takes are identified for each dimensions [48]. For each dimension, various shapes it can takes are identified. After then, different combinations of shapes from each dimensions are examined in order to find the practical solution to the problem. Although this methodology is based on the predefined structure of the solution spaces, it can facilitate the ideation process by helping to discover new relationships or configurations, which may not be so evident or might be overlooked by other [49].

At first, the fundamental dimensions of the business model were identified. These dimensions were derived from Osterwalder and Pigneur [50]'s Business Canvas, which is one of the most popular framework for describing the business model. Each perspective is defined as follows:

- Revenue streams: The way a company makes incomes from customer segments
- Cost structure: The cost that is necessary to operate business model and their structure
- Customer segments: Groups of a company tries to serve throughout providing value proposition
- Customer relationships: Types of relationship a company want to create with their customer segments
- Distribution channel: The path through which product or service travel from the company to customers
- Value propositions: the collection of product/service a business model offers to meet the needs of customers
- Key activities: The most important activities in creating and delivering a company's product/service
- Key resources: Resources that are necessary to execute key activities
- Key partnerships: External companies or Organizations which have a relationship with a company.

We adopt Osterwalder and Pigneur [50]'s scheme because it provides a balanced viewpoint for describing a business model by considering finance, customer and infrastructure aspects simultaneously.

For each business model dimension, a possible business modelling patterns were investigated. In this study, this elemental idea is called as 'business model strategy.' As previously mentioned, each strategy was identified by examining real business model cases. Figure 1 shows how business model strategies were identified. As illustrated in this figure, one of our collected cases was Netflix whose business model provides DVD through home delivery service instead of operating local shops. As a result, Netflix's idea was defined as 'delivery' strategy at distribution channel perspective. Another case, Apple, shows that a firm can generate additional profit by selling complementary product (music through iTunes) of installed-based product (iPod mp3 player). This strategy was defined as 'razor-blade' strategy at the revenue model perspective. Following same procedure, we have obtained total 69 strategies across the eight perspectives. The complete morphological chart of business model strategies is constructed and the definitions of each strategy and its corresponding business model case is illustrated in Table 1.

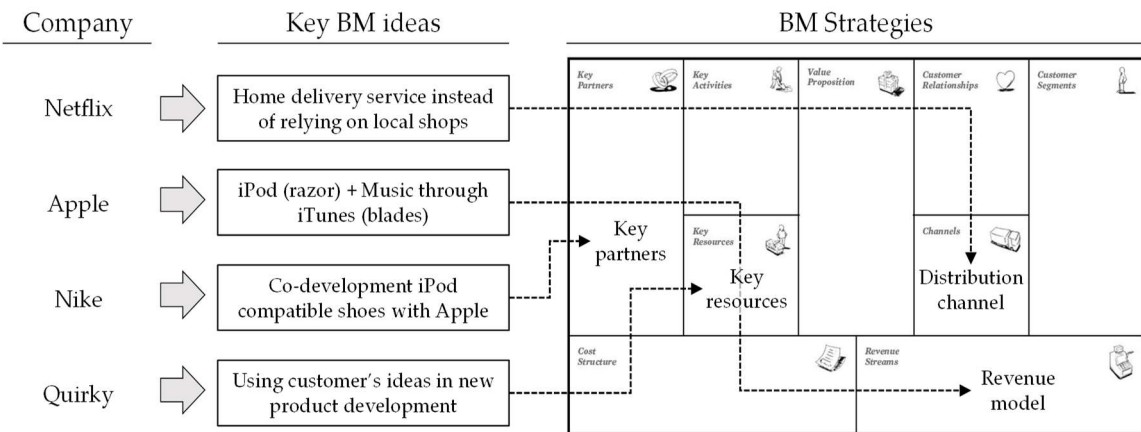

**Figure 1.** Example of business model strategy identification.

**Table 1.** Definition of the business model strategies (Morphological chart).

| BM Perspective | Strategy | Definition | Motivated Cases |
|---|---|---|---|
| Customer Relationships | Customer participation | allows customers to participate in value creation activities including design, production, delivery | Lego mind storm (allows customers to configure or design their own products) |
| | Reward | rewards customers according to improve customer loyalty | Pepsi (distributes Pepsi stuff point which can purchase variety of merchandise such as MP3) |
| | Upgrade | upgrade product/service after specific periods of product usage | Smartphone company (provides OS upgrade service) |
| | Blockbuster marketing | improves brand awareness throughout massive-scaled marketing event | Hyundai card (Credit card company hosts famous musician's concert) |
| | Life-cycle care | provides additional life-cycle service (maintenance, diagnosis, upgrade) after the product has been sold | Volvo (provides prognostic maintenance service for the vehicle) |
| | Customization | enables customers to purchase tailored product/service | Lutron lights (which are programmable so that customers can easily customize the aesthetic effect) |
| | Education | provides knowledge about efficient product/service usage | Lockheed martin (provides additional training service) |
| | Community | creates community where customer can communicate with other customers on product/service information | Turbotax (creates user community where customer can communicate about tax regulation or software usage) |
| | Viral marketing | induces word-of-mouth effect by encouraging customers to spread out their products/service | Groupon (induces customer to advertise specific product throughout SNS) |
| Distribution Channel | Experience shop | allows customers to experience product/services through distribution channel | Apple (allows customer to actively use products in their directly managed offline store, Apple store) |
| | Shop in shop | places small shop within big shop | Golden nail (sells coffees in nail shop) |
| | Delivery | delivers product/service to customers directly | Odin (daily vegetable/fruit delivery service) |
| | Sales person | uses sales calls in selling product/service | Woongjin Coway (water purifier company provides additional maintenance service through sales staffs) |
| | Road shop (Booth) | makes flexible/movable shops which can be placed in anywhere | Shamak (India movable laundry service) |
| | Web (mobile) | allows customers purchase product/service through internet connection | Bluenile (online jewellery shop) |
| | Indirect channel | uses generic distributors or wholesaler which are not owned by the company | Dell (begins to sell products in Wall-mart) |
| | Direct channel | operates direct distribution channel owned and operated by the company | Apple (sells iPhone through Apple store) |
| | Channel sharing | uses existing-but-non-traditional infrastructure as distribution channel | Seven eleven (enables customers to pick-up their packages at store) |

**Table 1.** *Cont.*

| BM Perspective | Strategy | Definition | Motivated Cases |
|---|---|---|---|
| Revenue Streams | Pay per unit | customer pays for single product/service | Variety of merchandise |
| | Subscription | customer pays fees for specific period in exchange of free usage of product/service | Netflix (provides subscription service for movie/TV contents) |
| | Razor blade | divides product into single platform (razor) and various complementary goods (blades) | Philips (separates coffee capsule from coffee maker, which is necessary continuously) |
| | Ad-based | customer see advertisement instead of paying fees | Tada copy (provides free copy service to students in exchange of putting ads on the other side of paper) |
| | Freemium | provides free version of product/service and receive fees when upgrading the product/service | Skype (free light version + premium version) |
| | Donation | donates certain amount of revenues for social values | Toms (donates some portion of shoes to African children) |
| | Commission | receives certain amount of revenues of product sold | Google |
| | Loyalty | receives loyalty fees from other parties | Technology-based companies |
| | Subsidiary | subsidy one party instead of receiving high fees from other part | Cell phone industry |
| | Pay as you want | enables customers to set their own fees | Radiohead (enables listeners to freely set their own price for downloading songs) |
| | Pay per use | customer pays fees as the exact amount of their usage of products/services | Zipcar (charges car rental fees according to customer's usage hours) |
| Customer Segments | Segment expansion | targets different customer segment | Petsmart (provides accommodation service to pets, not humans) |
| | Geographical expansion | targets different geographical region | Alibaba (enters the US market based on the success in the Chinese market) |
| | Long-tail targeting | sells a large number of unique items with relatively small quantities sold of each | Lovefilm (focuses on unpopular DVD contents) |
| | Premium targeting | targets product/service to high-end market | Louis Vuitton (focuses on a core trade, either a single product or a line of closely-related ones) |
| | Low-price targeting | targets sell product/service to low-end market | Southwest airline (provides low cost airline service) |
| | Social targeting | makes business model to improve social responsibility and sustainability | Grahmin bank (develops micro credit service to face with poverty) |
| | 2-sided targeting | targets more than two parties in order to exchange their needs | Innocentive (provides R&D brokerage services) |
| Key Resources | Recycle | uses waste product/materials in order to create new product | Threadless (recycles fabrics to make T-shirts) |
| | Crowdsourcing | uses an external resources (public, customer, amateur's) to create values | Quircky (crowdsourcing platform for new product/service development) |
| | Open source | promotes a partial/universal access to a design or modify the product | Google (develops open source-based modular cell phone and each of modules are accessible/modifiable to any external developers) |
| | Outsourcing | reduces costs by transferring portion of work to outside suppliers | Apple (outsources its manufacturing and supply chain part) |
| | Alliance | makes coalition or friendship between two or more companies | Google (develops partnership with Samsung to develop Nexus) |
| | Brand leverage | uses the power of an existing brand name to support a company's entry into new but related product category | Nespresso (uses Nestlé's brand equity in launching espresso machine) |
| Key Activities | Added service | servitizes existing business process | GE (adds engine maintenance service) |
| | Service productization | productize/automates existing business process | Redbox (provides DVD rental service through kiosk) |
| | Standardization | lowers cost by standardizing business process | McDonalds (highly standardizes menus and recipe) |
| | Economies of scale | makes use of cost advantages due to scale of operation | Wal-Mart (can buy enormous bulk, force suppliers to accept low price and sells at low price to customers) |
| | Economies of scope | makes use of efficiencies by variety of product/service options | Sony (sells large variety of Walkman series) |
| | No frill | non-essential features have been removed to keep price low | EasyJet (cuts down the aircraft fees by eliminating unnecessary in-flight service features) |

**Table 1.** *Cont.*

| BM Perspective | Strategy | Definition | Motivated Cases |
|---|---|---|---|
| | Self service | reduces cost by allowing customers to complete most steps in purchasing or using of product/service | Velov (bike sharing system with unmanned station) |
| | Peer to peer | enables individuals to distribute, share and reuse of their excessive capacity with other individuals | Airbnb (website for people to rent out lodging) |
| | R&D collaboration | involves partners in technology development | Coca-Cola + Heinz (collaboration to develop more sustainable containers) |
| | Design collaboration | involves partners in design phase | Nike (develops iTunes-compatible health signal tracking shoes) |
| Key Partnerships | Joint distribution | makes use of partner's distribution channels | Netflix (installs DVD rental shop at Walmart) |
| | Shared investment | makes co-investment to share risk | Repsol + Burger King (collaboration to increase revenue at gas stations) |
| | Cross promotion | promotes each other's product/service | Mallskin (develops a diary book which is compatible with Evernote) |

As shown with Table 1, our morphological chart identifies about five to ten strategies for each business model perspective. By examining various combinations of business model strategies, one can generate various business model ideas at PSS planning stage. Also, it is noteworthy that the proposed morphological chart could be continuously updated as new business model strategies are identified.

## 4. Case Application: PSS Business Modelling of Hair Dryer Company in South Korea

### 4.1. Current Buiness Model

This chapter illustrates how our morphological chart can be used in actual business modelling. Let us take an example of hair dryer manufacturer who actually used our morphological chart in planning their new business model. This company is one of the largest hair dryer manufacturer in South Korea. However, recent growth of Chinese manufacturers made the company to compete with fierce price competition. Moreover, due to the fairly long life-cycle of hairdryers, there is little potential for generating additional profit from existing customers. Therefore, the company try to develop a new PSS concept in order to differentiate their business model from competitor's in order to avoid price competition and generate new values for customers.

In order to find their new business model, a series of workshops were held where our morphological chart was introduced and intensively used in business modelling process. At first, the company's current business model was analysed with morphological chart. Their major product line have mainly targeted home users. Accordingly, the product has simple function with low price and were distributed through home appliance stores. The combination of strategies of existing business model is shown in Figure 2. As can be seen, the 'sell-per-unit' is chosen as a revenue streams strategy because the firm charges each hair dryer with fixed prices. For the distribution channel perspective, the 'indirect channel' strategy is chosen because they do not operate their own channels. Finally, because they focused on gaining price advantages with large sales volumes, the 'economies of scale' are chosen as their strategies of activity configurations. The remaining perspectives remains empty because no special strategies are identified.

Based on their current business model, all possible strategies for each perspective are examined in order to find meaningful alternatives. With detail description about each alternative, their representation model is as follows.

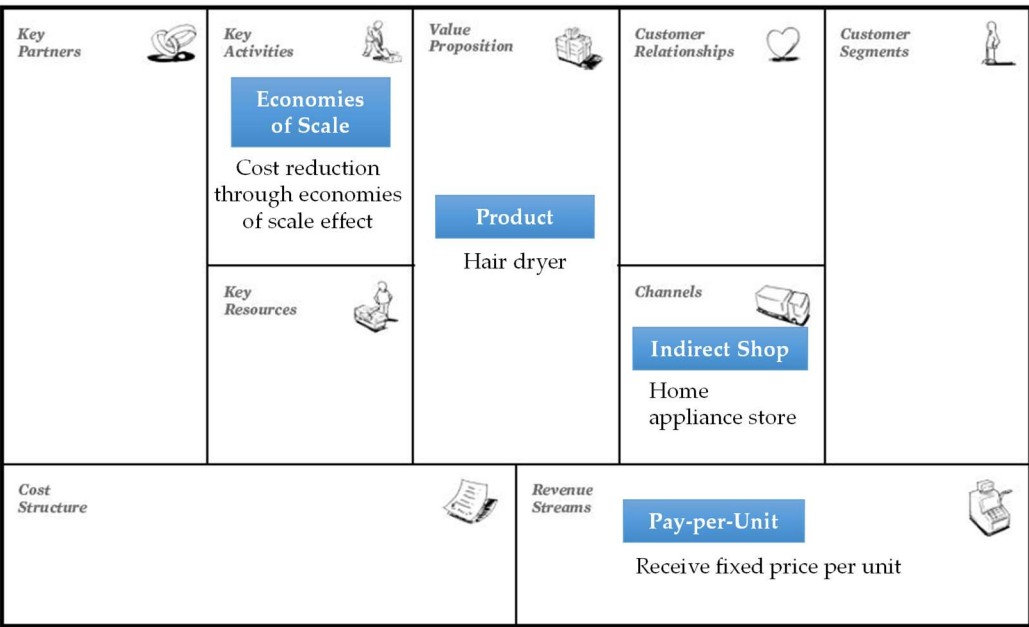

**Figure 2.** Strategy analysis of current business model.

## 4.2. Alternative #1: Razor-Blade Business Model

One limitation of current business model was relatively small size of hair dryer market. Although the company has highest market share in the segment, the sales volume begins to decrease due to fairly long life-cycle and market saturation. Therefore, creating a new sustainable revenue stream was their major concern. In this sense, the 'razor-blade' strategy at revenue model is examined. This strategy aims to separate install-based product and complementary good in order to generate additional profit from complementary goods. Motivated by this strategy, they proposed the idea of ampule-compatible hair dryer. The proposed ampule consists of nutrient which can keeps hair and scalp with healthy state. When the ampule is installed with hair dryer, the nutrient spreads throughout winds generated from the hair dryers. With this strategy, the firm can expect continuous revenue streams by selling ampules to hair-dryer owners.

## 4.3. Alternative #2: Partnership with Chemical Company

Although the first alternative can generate additional revenue streams, the manufacturer has no capability of developing hair-dryer-compatible ampules with their own. One solution to tackle this issue is to develop a partnership with other companies. As a result, one of partnership strategy, the 'design collaboration,' was examined. Accordingly, as a product development partners, the manufacturer can involve famous chemical company which has been famous in hair chemical products. Adopting this strategy also can lead to the 'brand leverage' effect than developing the chemical product their own. The chosen business model strategy is described in Figure 3.

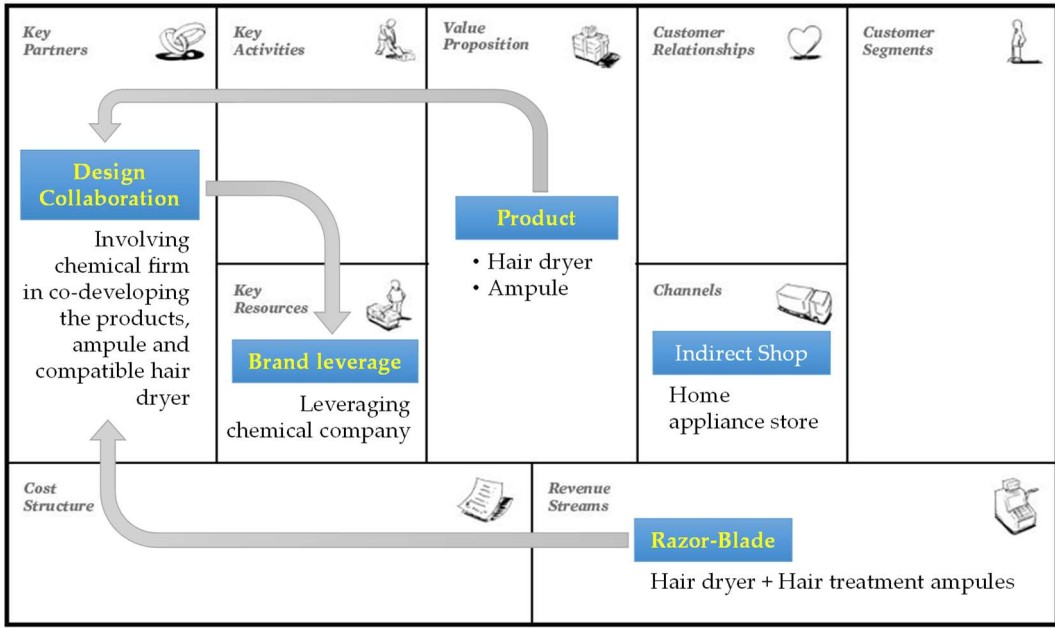

**Figure 3.** Strategy analysis of the business model alternatives #1 and #2.

### 4.4. Alternative #3: Targeting Professional Markets (B2B)

The third idea was motivated by the 'segment extension' strategy in the customer segmentation perspective. Until now, the manufacturer has only focused on home appliance markets. Instead, ampule-compatible hair dryer may be better used for professional market which consists of hair-shop owners. This idea is more viable because there is little chance that home users continuously use costly the ampules. Furthermore, hair-shops also serve as a good distribution channel for the manufacturer. New customers who are not conscious of the hair dryer and the ampules would have a good chance to experience the product/services in the hair shop. Therefore, the 'channel sharing' strategy is invoked at distribution channel perspective. Motivated by this strategy, it would be proposed that hair-shops sell the hair dryer and the ampules to customers who shows an interest.

### 4.5. Alternative #4: Developing New Services Packages

In this scenario, the 'added service' strategy at key activities perspective, which servitizes their existing business process, is adopted. That means this business model alternative is motivated with developing a hair-treatment service package which utilizes the ampule-compatible hair dryers. Similar with previous alternatives, the hair-shops become good distribution channels for the product/service provision. In this business model, however, the firm aims to provide more professional service packages including diagnosis of scalp and hair and to customize the recipe of ampules for each customer. Furthermore, the 'education' strategy, training the proper use of the product/service, is required in order to offer better professional hair-treatment service. The resulting business model is depicted in Figure 4.

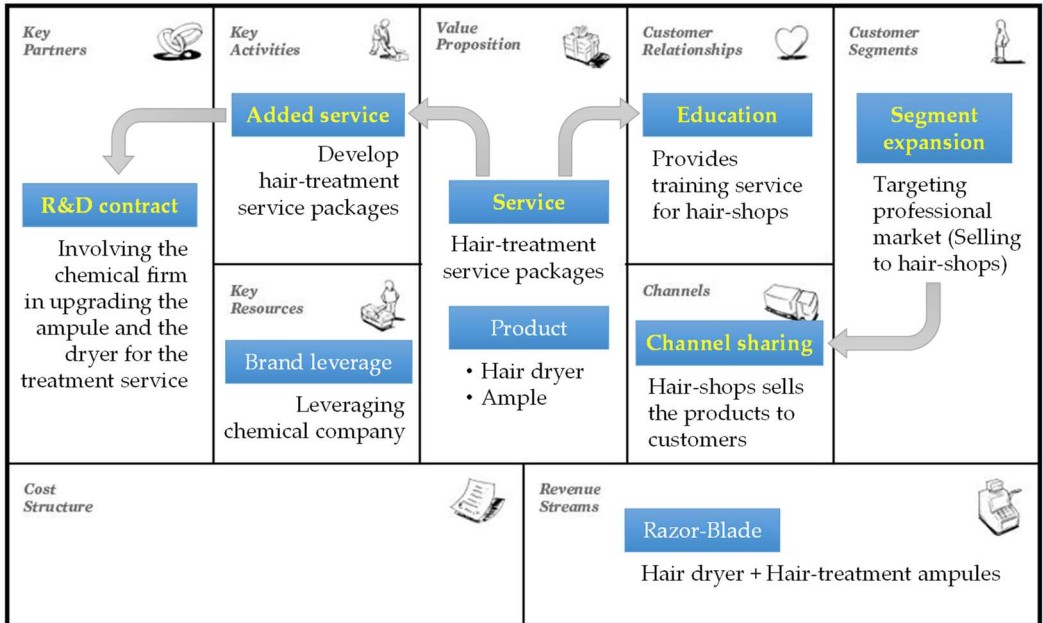

**Figure 4.** Strategy analysis of the business model alternatives #3 and #4.

### 4.6. Alternative #5: Productizing the Service Packages

The final alternative is to productize the hair-treatment service, which adopts the 'productization' strategy from the key activities perspective. It aims to develop a product which can automate the service activities performed by humans. Accordingly, a new device for the self-doing the hair-treatment service package, which diagnoses scalp and hair and create a customized care recipe. Moreover, the 'segment extension' strategy, from the professional users to the unprofessional users, is again adopted, because this device is intended for home appliance markets

Finally wrapping up the business model alternative above, the business model roadmap which present the innovation path among the business model alternatives is developed (Figure 5). Each business model alternative is mapped along with the degree of servitization, product-service ratio and the time table, degree of business model evolution. Therefore, the firm could have generated a plan when and how to implement each business model.

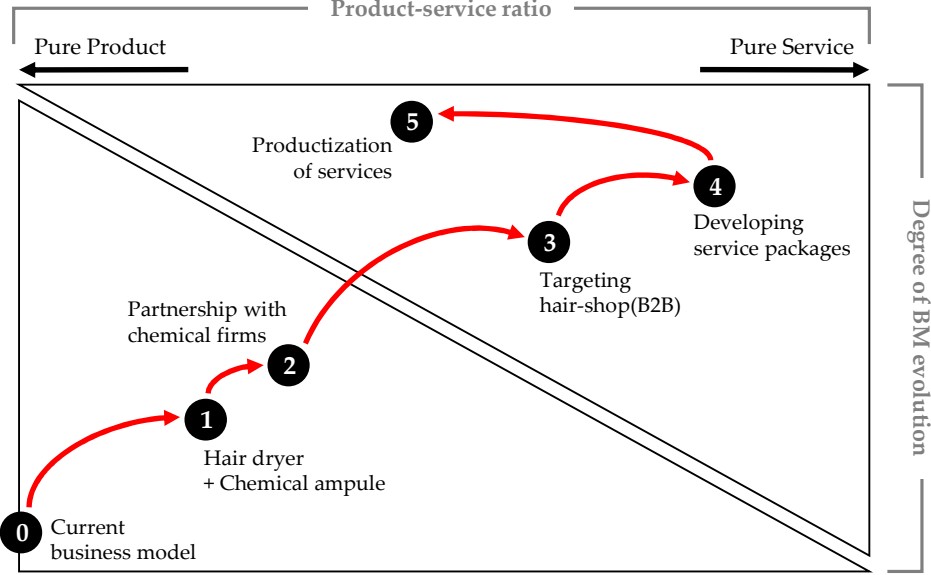

**Figure 5.** Roadmap of business model innovations.

## 5. BizChef: PSS Business Modelling Support System

In order to facilitate the practical use of the proposed morphological chart, the prototype system, named BizChef, was developed [51]. The main objective of this software is to enable a user to explore various business model cases and reuse these cases in generating a new business model ideation. In terms of supporting PSS design, a computer-aided design system for PSS have proposed by the literature [52–54]. Their system mainly focuses on the detail design of artefact and service process in order to provide a functions that meets the PSS customer requirements. Contrary to this, our software focuses on the planning of PSS business models at the strategic level.

The basic architecture of BizChef is illustrated in Figure 6 and it consists of one BM database and two function modules; BM Exploration and BM Creation. In the BM database, real business model cases are analysed through the morphological chart. Those cases are collected by searching numerous sources including case study article, annual report or website of relevant companies. Each case is then classified according to business model strategies defined over morphological chart. Total 155 business model cases are accumulated in the current version of database. Based on this database, the two functions are provided.

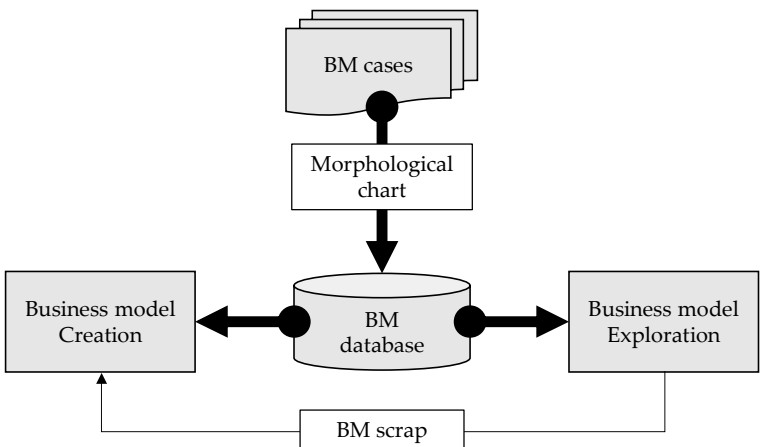

**Figure 6.** Architecture of the BizChef.

Business model exploration module enables users to search various business model cases. There are two approaches for more efficient search; searching by industry categories (Figure 7a) and searching by the strategies (Figure 7b). If users choose an industry category or a strategy as an input query, a list of relevant business model cases. If users select a specific business model case among the list, detail description and pictures for the case (Figure 7c) as well as the strategies for each perspectives and the reason why each strategy (Figure 7d) is adopted are presented. User can also scrap interested business model cases and strategies (Figure 7d). These scrapped strategies can later be used in creating a new business model.

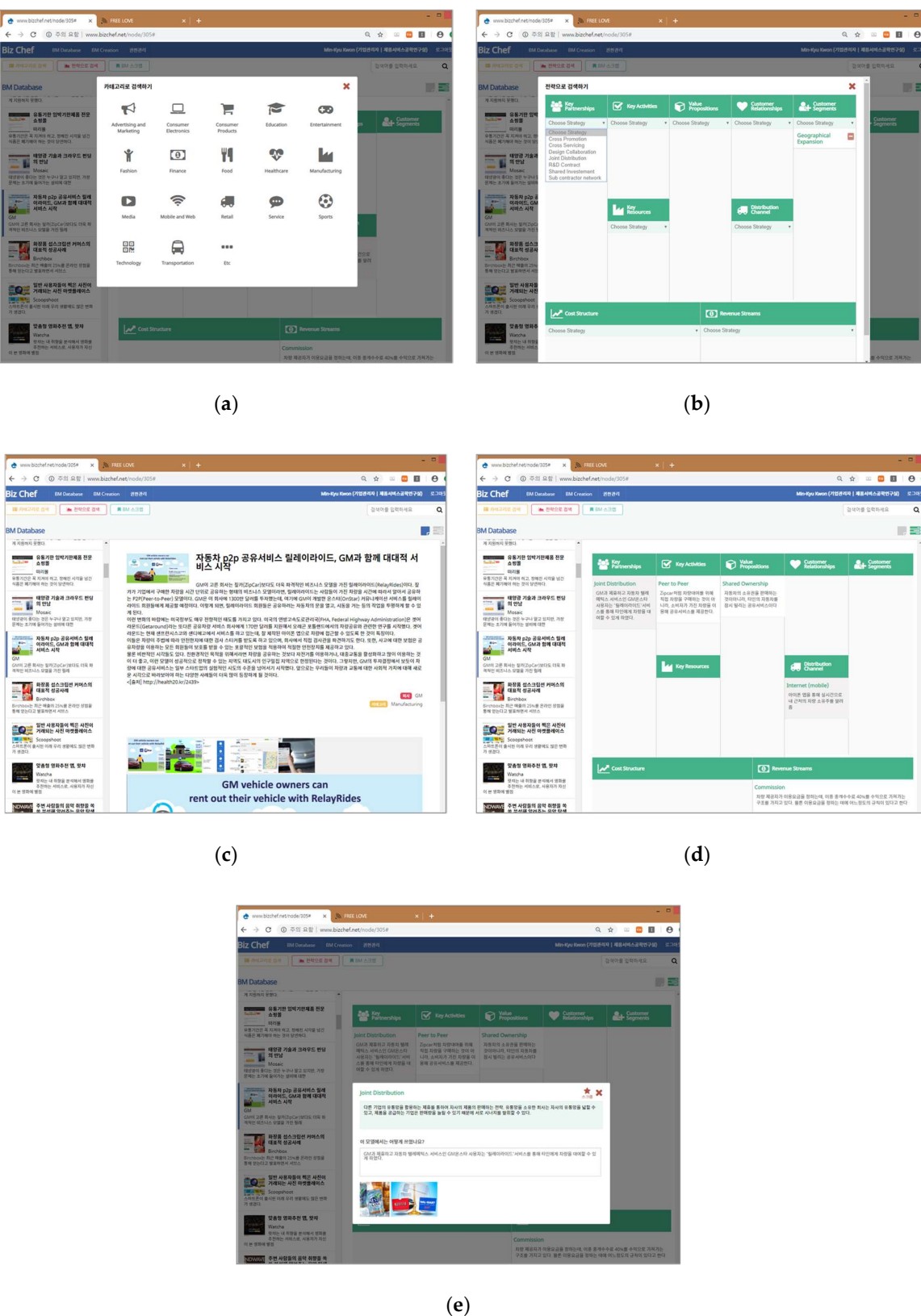

**Figure 7.** (**a**) BM case search by an industry category; (**b**) BM case search by a business model strategy; (**c**) Description of a BM case; (**d**) Implemented strategies of a BM case; (**e**) Scrap of a strategy in a BM case.

Business model creation module enables users to systematically generate and manage their own business models. Figure 8 shows a snapshot of business model creation module; describing details of a business model creation project (Figure 8a) and editing strategies from each perspectives for the project (Figure 8b). In business model creation projects, users can reuse strategies which have been scrapped from the exploration module.

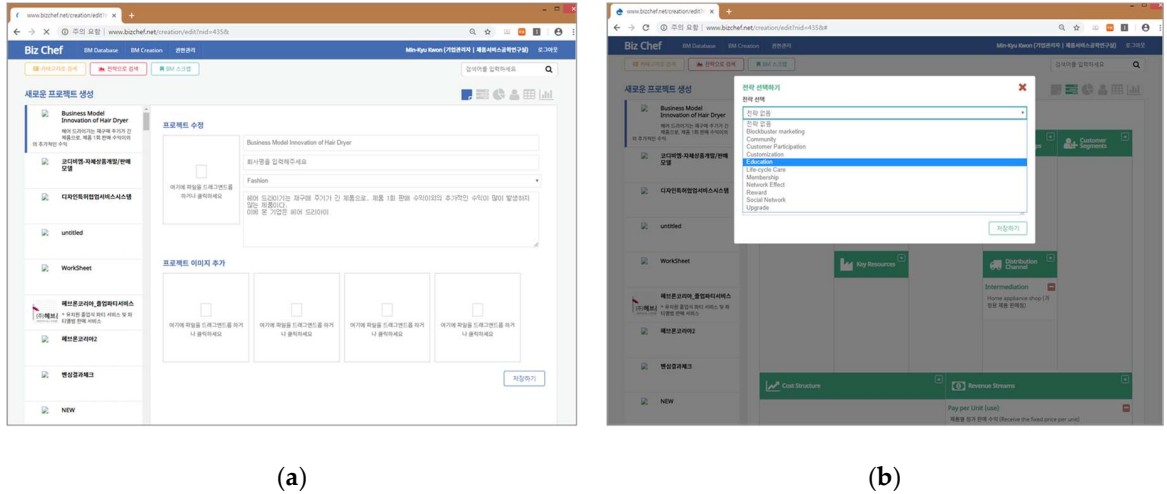

(**a**)                                                                 (**b**)

**Figure 8.** (**a**) Description of the business model creation project; (**b**) Adoption of a business model strategy for each business model perspective.

## 6. Conclusions and Future Works

Despite of its importance, relatively little attention has been made to business model planning in PSS literature. Present study introduces a procedure for generating various PSS business model ideas. To provide a systematic guideline for business model idea exploration, a morphological analysis was conducted to identify all the possible business modelling ideas. Each idea was derived from in-depth examination of real business model cases. One can enhance ideation process by mixing and matching pre-defined building blocks. From real case study of hair dryer manufacturer, we have demonstrated the modelling benefit of business modelling using proposed morphological chart. Moreover, a case-based system for supporting business modelling, BizChef, was introduced. This system consists of database wherein a more than hundreds of real business model cases are analysed throughout morphological chart. Based on this database, our system enables a user to efficiently explore business model cases and use them in actual business modelling process.

There are, however, further research issues to explore. Although we have less than 10 strategies for each perspectives of a business model, examining all the possible combinations of them may not be practical. One way to reduce this number is to find inter-relationship between strategies. For example, if strategies are grouped by their similarity or forbidden combinations of strategies are identified in advance, we can narrow down entire solution spaces more efficiently. Therefore, one of our future research is to find such interaction between strategies. Another issue is development of business mode evaluation method. Currently, we have focused on idea generation process. However, in order to compare numerous competing business model alternatives, an evaluation method for business model is also required. In order to develop this method, identifying a balanced set of evaluation criteria should be examined.

**Author Contributions:** All the authors contributed equally to all aspects of the research presented in this paper. M.K. developed the BizChef system; M.K. and J.L. collected, analysed the business model cases and wrote the manuscript; J.L. developed the research framework; Y.S.H. conceptualized and supervised the research project; all the authors have reviewed and edited the manuscript.

**Funding:** This research was funded by National Research Foundation of Korea(NRF) funded by the Ministry of Education (NRF-2017R1C1B5076752). This work was partially supported by the "Core Technology Development Program for Knowledge-Based Service" funded by the Korean Government (MOTIE) (Project No. 10048090, Title: Manufacturing Servitization Support Framework). This research was also financially supported by the Ministry of Trade, Industry and Energy (MOTIE) and Korea Institute for Advancement of Technology (KIAT) through the National Innovation Cluster R&D program (Project No. P0006910, Title: Development of recommendation business system based supplies with blockchain).

**Conflicts of Interest:** The authors declare no conflict of interest.

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
