# Peer review of "Product-Service System Business Modelling Methodology Using Morphological Analysis"

_sustainability, doi:10.3390/su11051376_

Round 1

Reviewer 1 Report

The paper describes the development of a methodology for PSS business modelling by means of the morphological reasoning approach. The validity of the study is discussed through a case study where the proposed approach is applied to a hair drier company. Finally, the authors propose a supporting SW that makes the use of the proposed approach easier, allowing an exploration of various business model cases that can be reused to generate new business models.

I think that these issues appear relevant in context of PSS development as a means for the implementation of more sustainable solutions.

However, to become suitable for publication, the manuscript needs to be improved, especially augmenting its scientific quality.

My suggestions mainly concern following points:

In the introduction, the research question and targets should be clearer. The Xerox example (lines 27ff) could be omitted, while more referenced information should be inserted supporting the research motivation (for example, you might consider the following study: https://doi.org/10.1002/bse.1939 ) and the proposed approach. For instance, at lines 50ff, the morphological approach is introduced without any reference. Accordingly, also the expression “structured methodology”, which is used several times in the text, needs to be justified or replaced by a simpler definition, e.g. “procedure”.

The background analysis appears quite partial, as the use of the morphological approach is not a novelty in the PSS development context. For these reasons, it is recommended to mention at least the following works of Dr. Haber:

2. DOI: 10.1504/IJPD.2017.086474

This can allow a more comprehensive analysis, supporting the potential benefits of the proposed procedure.

In section 3, the expression “traditional approach” (line 194) needs to be explained; accordingly, the sentence “To  that  end,  the  present  study  introduces comprehensive methodologies and a framework for designing a new business model” (lines 313-314) needs to be revised. As a reference you could consider the following paper, where a definition of methodology in engineering design is provided taking into account the use of the morphological matrix/chart as well: https://doi.org/10.1016/j.jengtecman.2013.09.005.

Finally, when considering the output of your study, a reference to studies concerning existing SW for the implementation of PSS models should be made, comparing them: e.g.  https://doi.org/10.1016/j.compind.2012.02.009; https://doi.org/10.1080/09544820903151715; https://doi.org/10.1080/09537287.2015.1033493.

Besides, the language should be improved in order to make easier going through the manuscript. For example, Line 41-42 “… literature attempted to clarify the concept of the business model with an academic viewpoint”. This sentence is quite tautological.

Author Response

We thank the review team for supporting our paper and for the constructive criticism. We have received excellent comments that are valuable in improving the manuscript. We are pleased to report that we were able to successfully reflect all the concerns of our reviewers to the revised manuscript. In this document, we explain how we respond to the comments that we have received.

Reviewer 2 Report

The paper proposes a methodology for structured business modelling, aiming at successful implementation of product-service systems (PPS). By using morphological analysis, different patterns of the PSS business model are identified, and used as reference through the morphological chart in the innovative business model design process. Moreover, a case study of a hair dryer company is also presented. The proposed business model is generic and it can be applied to various contexts. The morphological chart with 69 business strategies is a very valuable tool developed by the authors.

Please consider the following remarks to improve the paper.

1.      Line 49: double “to”

2.      As the morphological analysis, based on the morphological matrix, is a well-known approach in conceptual design (and not only), I suggest to avoid the syntagm “our methodology” (see line 16, etc.), but you can refer to the business model patterns identification and their compatible combinations, or to the morphological chart of business strategies.

3.      Please check carefully the English grammar used in the paper, there are several grammatical mistakes.

4.      Line 303: Figure 9 is missing.

Author Response

(The authors gave the same response as above.)
